# Osteonecrosis in Korean Paediatric and Young Adults with Acute Lymphoblastic Leukaemia or Lymphoblastic Lymphoma: A Nationwide Epidemiological Study

**DOI:** 10.3390/jcm11092489

**Published:** 2022-04-28

**Authors:** Seung Min Hahn, Myeongjee Lee, Aaron Huser, Yeonji Gim, Eun Hwa Kim, Minsoo Kim, Amaal M. Aldosari, Inkyung Jung, Yoon Hae Kwak

**Affiliations:** 1Division of Pediatric Hematology & Oncology, Department of Pediatrics, Severance Hospital, Yonsei University College of Medicine, Seoul 03722, Korea; bluenile88@yuhs.ac; 2Biostatistics Collaboration Unit, Department of Biomedical Systems Informatics, Yonsei University College of Medicine, Seoul 03722, Korea; mlee1004@yuhs.ac (M.L.); ehkim0607@yuhs.ac (E.H.K.); 3Paley Institute, West Palm Beach, FL 33407, USA; aaron.huser@gmail.com; 4Asan Medical Center Children’s Hospital, Department of Orthopaedic Surgery, College of Medicine, University of Ulsan, Seoul 05505, Korea; yeonji@yuhs.ac (Y.G.); minsoo1p@naver.com (M.K.); dr_amaal@hotmail.com (A.M.A.); 5Department of Orthopaedic Surgery, Al Noor Specialist Hospital, Makkah 24242, Saudi Arabia; 6Division of Biostatistics, Department of Biomedical Systems Informatics, Yonsei University College of Medicine, Seoul 03722, Korea

**Keywords:** osteonecrosis, acute lymphoblastic leukemia, lymphoblastic lymphoma, adolescent and young adult, Republic of Korea

## Abstract

Osteonecrosis (ON) is a serious complication of acute lymphocytic leukaemia (ALL) or lymphoblastic lymphoma (LBL) treatment, and there is little information regarding ON in Korean paediatric and young adult patients. This retrospective cohort study assessed the cumulative incidence of and risk factors for ON using national health insurance claims data from 2008 to 2019 in 4861 ALL/LBL patients. The Kaplan–Meier method was used to estimate the cumulative incidence of ON according to age groups; the Cox proportional hazard regression model was used to identify risk factors related to ON development after diagnosing ALL/LBL. A cause-specific hazard model with time-varying covariates was used to assess the effects of risk factors. Overall, 158 (3.25%) patients were diagnosed with ON, among whom 23 underwent orthopaedic surgeries. Older age, radiotherapy (HR = 2.62, 95% confidence interval (CI) 1.87–3.66), HSCT (HR = 2.40, 95% CI 1.74–3.31), steroid use and anthracycline use (HR = 2.76, CI 1.85–4.14) were related to ON in the univariate analysis. In the multivariate analysis, age and steroid and asparaginase use (HR = 1.99, CI 1.30–3.06) were factors associated with ON. These results suggest that Korean patients with ALL/LBL who used steroids and asparaginase should be closely monitored during follow-up, even among young adult patients.

## 1. Introduction

Significantly improved survival rates of >90% in children and adolescents with acute lymphoblastic leukaemia (ALL) or lymphoblastic lymphoma (LBL) indicate treatment strategy concerns of long-term side effects of treatments and disease remission [1,2]. Osteonecrosis (ON) is a common and serious side effect that compromises quality of life (QOL) [3,4]. ON incidence varies widely (1.6–17.6%) depending on the study cohort and treatment protocol [5,6,7,8,9,10,11,12,13,14,15]. ON aetiology is multifactorial, with risk factors such as steroid use, sex, body mass index (BMI) [16], genetics [2,7,10,17,18], haematopoietic stem cell transplantation (HSCT) [19] and antileukaemic agents [1,3,11,19,20,21,22,23,24,25,26]. Despite many studies, the results remain debatable [1,21,27], although adolescents aged 10 years and older have the highest risk of developing ON [3,12,22,24,28]. Steroids are essential in the induction phase for ALL treatments; however, high doses of steroids are one of the most important treatment-related risk factors [24,29].

ON treatment slows disease progression and the collapse of major joints, and leads to pain relief and increased range of motion; however, there is no standard treatment for ALL-associated ON [16,29,30,31,32,33]. Some patients complain of symptoms after progression [1,34,35,36]. ON usually occurs in major joints where injury to cartilage is difficult to revitalise [30]. Early diagnosis and prevention of ON are important for preserving QOL after remission in children with malignancies [37]; furthermore, definitive risk factors remain unknown, although some factors such as steroids and antileukaemic agents are critical for survival and remission. There have been prospective randomised control studies and multicentre studies on ON after ALL/LBL treatment [19,26,38,39]; however, there have been no large-scale clinical studies on Korean patients. Because genetic and demographic differences are possible risk factors, we evaluated ON incidence, possible risk factors, and rate of orthopaedic surgery among patients with ALL/LBL based on a national cohort with the Korean National Health Insurance claims database.

## 2. Materials and Methods

### 2.1. Data Source

This study was an interdisciplinary (haemato-oncology/orthopaedics/biostatics), nationwide, retrospective, cohort study using the Korean Health Insurance and Review Assessment (HIRA) database. The HIRA database comprises nationwide health insurance claims and covers approximately 98% of the total population; it contains beneficiaries’ general characteristics and also all diagnoses, procedures, treatments and inpatient and outpatient prescriptions [40]. In the HIRA database, all beneficiaries’ and healthcare providers’ identification are encrypted according to the Health Insurance Portability and Accountability Act privacy rules, and diagnoses are coded using the International Classification of Diseases 10th Revision. Both inpatient and outpatient claims from the 2008–2019 database were used. This study was approved by the Care Record Supply Committee of the HIRA (M20210201983) and the Institutional Review Board/Ethics Committee of Severance Hospital, Yonsei University Health System (IRB no. 4-2019-1017), which also waived the need for informed patient consent because of the retrospective study design and use of de-identified data.

### 2.2. Study Population

This study included individuals aged <40 years diagnosed with ALL/LBL between 2008 and 2019 (*n* = 8032). Patients with ALL and LBL were defined using ICD-10 codes C91 and C83.5, respectively, with codes for expanding benefit coverage (V-code in Korea). Policies to expand benefit coverage commenced in 2005 and have reduced medical expenses for catastrophic illnesses, including cancer and cardio-cerebrovascular diseases [41]. A 1-year washout period was applied to exclude prevalent cases (*n* = 2967) (Figure 1). The first ON diagnosis after ALL/LBL diagnosis was the event of interest. ON was defined using ICD-10 code, M87 or M90. Patients with ALL/LBL diagnosed with ON within 30 days after ALL/LBL diagnosis were excluded because they could have ON regardless of ALL/LBL treatment (*n* = 31). Moreover, we excluded 82 patients with ALL/LBL who did not receive anticancer therapy during the follow-up period. Finally, 4861 patients with ALL/LBL were identified from the database.

### 2.3. Possible on Risk Factors

Participants were divided into the paediatric group (<20 years), which was further subdivided into five-year increments and the young adult group (≥20 years). Radiotherapy treatments included total-body and craniospinal irradiation. HSCT included autologous and allogeneic transplants. Steroid usage was analysed to compare different types of steroids used during induction. We reviewed steroid use in the first 60 days after diagnosing ALL/LBL. The dexamethasone group included patients receiving dexamethasone during induction treatment. The prednisone group used prednisone, the dexamethasone + prednisone group used both, and the no steroid group did not use dexamethasone or prednisone during the early treatment phase. L-asparaginase (ASP) is a key drug in the paediatric or paediatric-inspired ALL/LBL regimen; therefore, we identified patients who used L-ASP at least once during the entire treatment period or who never used L-ASP. The use of anthracycline (daunorubicin or doxorubicin) was also analysed during the first 60 days. Anthracycline drugs are usually included in the first-line treatment of adult ALL/LBL and are primarily used in high-risk paediatric ALL during induction.

### 2.4. Statistical Analysis

Baseline characteristics of the study population are summarised using mean and standard deviation for continuous variables and frequency and percentage for categorical variables. Comparisons between the two groups were conducted using Wilcoxon rank sum test for continuous variables and chi-square tests for categorical variables. Patients with ALL/LBL were followed from the diagnosis date until the date of ON diagnosis (event of interest), death, or end of the study (31 July 2019), whichever came first. The Gray’s method was used to estimate the cumulative incidence (Cuml) of ON according to the age group. A cause-specific hazard regression model was fitted with a Cox proportional hazard regression model to identify risk factors that could affect ON development after ALL diagnosis. Age at ALL/LBL diagnosis, sex, radiotherapy, HSCT, steroid use, ASP use and anthracycline use were considered possible risk factors. Radiotherapy and HSCT were considered time-varying risk factors. Univariate and multivariate analyses were performed to assess the effect of a risk factor without and with adjusting for the effects of other risk factors. The results are presented as hazard ratios (HRs) with 95% confidence intervals (CIs). Statistical significance was defined as two-sided *p*-values < 0.05. SAS Enterprise Guide version 9.4 (SAS Institute Inc., Cary, NC, USA) and R 3.6.1 (R Foundation for Statistical Computing, Vienna, Austria) were used for all statistical analyses.

## 3. Results

### 3.1. Participants’ Characteristics

Among 4861 participants diagnosed with ALL/LBL, 3344 (68.8%) were aged <20 years. The mean age of all participants was 14.9 years; the paediatric group, 8.3 years, and the young adult group, 29.3 years. Male subjects were more common (2885, 58.4%) in all age groups. Radiotherapy was administered to 995 (20.5%) patients and was the most common among adults (34.3%). HSCT was administered to 54.9% of patients aged >20 years compared with 16.8% in the paediatric group. Prednisone was the most frequently prescribed steroid in the paediatric group (49.8%) during the first 60 days after diagnosis, whereas dexamethasone was predominant in the adult group (37.4%). For patients aged <20 years, prednisone + dexamethasone was used in 23.4%. Neither prednisone nor dexamethasone was used for 10.6% of patients aged <20 years and 12.4% of adults. L-ASP was used in 81.8% of patients aged <20 years and 40.5% of adults. Daunorubicin or doxorubicin was administered to 3149 (64.8%) participants in the early phase of ALL/LBL treatment and was more frequently used in the paediatric group than in the young adult group (*p* < 0.01). The paediatric group used anthracycline more frequently than the young adult group (1940 vs. 308, *p* < 0.01) (Table 1).

### 3.2. Incidence of ON

ON was diagnosed in 158 (3.25%) of 4861 patients with ALL/LBL, with 93 aged <20 years, i.e., 58.9% of all cases, and 65 patients aged ≥ 20 years. The mean age at ALL/LBL diagnosis was 19.41 years for patients who developed ON and 14.73 years for those who did not. In the paediatric group, the mean age at ALL/LBL diagnosis was 13.28 years for patients who developed ON and 8.20 years for those without ON (Figure 2). ON occurred more frequently in the young adult group than in the paediatric group (4.28% vs. 2.78%). However, the Cuml of ON at 10 years was the highest in patients aged 10–20 years (7.34%), especially in the age range of 15–20 years (8.38%) (Figure 3). The mean age at ON diagnosis was 22.0 ± 8.81 years, which is 2.54 ± 1.74 years after ALL/LBL diagnosis in all patients. Time from cancer treatment to ON diagnosis was similar in the paediatric (2.69 ± 1.77 years) and adult groups (2.33 ± 1.68 years) (*p* = 0.20). Approximately 90% of patients with ON were diagnosed within 5 years of the primary cancer (Figure 4).

### 3.3. Risk Factors for ON

Age, radiotherapy, HSCT and anthracycline use in the first 60 days after ALL/LBL diagnosis were associated with ON among all patients with ALL/LBL (Table 2). This was evident among paediatric patients, however, these factors were not statistically significantly related to the occurrence of ON in the young adult group (Appendix A). In the cause-specific regression model, age, radiotherapy, HSCT, steroid use and anthracycline use were significantly associated with ON risk among all patients with ALL/LBL in the univariate analyses. In the multivariate analysis, older age, steroid use and history of L-ASP use were statistically significant risk factors for ON development. Male patients had a 26% lower risk for ON than female patients, although it was not significant (HR = 0.74, 95% CI 0.54–1.01, *p* = 0.06), this effect was significant in paediatric patients (*p* < 0.001) (Table 3).

In the regression analysis with cause-specific HRs and time-varying covariates, older age, radiotherapy, transplantation, steroid use and anthracycline use were related to ON diagnosis in the univariate analysis. In the multivariate analysis, age, steroid use and history of L-ASP use were factors associated with ON. Males showed lower rates of ON than females, and radiotherapy and transplantation were not associated with ON in the multivariate analysis. Analysis of the paediatric age group showed similar results. Age, steroid use and L-ASP use were related to ON in the multivariate analysis, whereas in the adult group (20–39 years), age, sex, radiotherapy and transplantation were not associated with ON in the univariate analysis (Table 3). No risk factors were statistically significant in the development of ON in young adult ALL/LBL patients from multivariate analysis (Appendix A).

### 3.4. Surgical Treatment of ON

Among 158 patients with ON, 23 (14.5%) underwent orthopaedic surgery; 21 patients had ON at the hip joint and 1 knee and 1 shoulder. Among these, most (87%, 20/23) underwent total hip arthroplasty (THR). Three patients underwent arthroscopic debridement: one hip, one knee and one shoulder. The mean age of patients who underwent THR was 28.1 years, with two aged <20 years. An MRI was performed for 48 patients (30.4%, 48/158), and eight patients received surgical treatment (34.8%, 8/23).

## 4. Discussion

To our knowledge, this is the first study to present the Cuml and risk factors for ON after ALL/LBL treatment in a national cohort comprising Koreans aged <39 years with several important findings.

First, ON was associated with older age in the paediatric group, and young adults (20–39 years) had a higher ON incidence (4.28%) than paediatric patients (2.78%). ON is a critical complication for the QOL of young adults after antileukaemic therapy; however, relatively few studies have been conducted in these age groups [42]. Our study, like others, showed that Cuml is the highest for patients aged 15–20 years, but the very low incidence in those aged <10 years resulted in a lower Cuml in the paediatric group overall than in the young adult group. These results are inconsistent with those of previous studies [43,44,45]. As increasing numbers of young adults receive a paediatric protocol rather than adult protocols with a concomitant higher dose of steroids, our study showed a high incidence in the young adult group [39,46]. Heneghan et al. reported 244/10,729 (2.33%) patients with ALL, which is consistent with our study [47]. Although paediatric-inspired protocol use in adults is increasing, the use of dexamethasone in adults remains higher than that in the paediatric group (37.38% vs. 16.27%, Table 1), which could be related to the relatively high incidence in adults. Second, sex is considered an important risk factor, but there are inconsistent reports [43,44,45]. In our study, female was a risk factor only in the multivariate analysis in the paediatric age group (male, HR = 0.58, *p* < 0.01). There remains controversy regarding females, and the literature describes that an increase in oestrogen levels in females could have procoagulant effects, but our results suggest that differences in puberty between females and males may contribute to ON development [3,27,48]. Third, the median time to develop ON was 2.54 years, with no difference between the paediatric and young adult groups (2.69 vs. 2.33 years, *p* = 0.20). This duration was longer than that reported in previous studies [35,45]. Patients in our study could be diagnosed based on progressive symptoms rather than anticipative evaluation or because only clinically symptomatic ON was recorded. Fourth, our study showed that radiotherapy and HSCT were risk factors in the univariate analysis. The subset of patients receiving these treatments is supposed to be exposed to total-body radiation and considerably higher doses of corticosteroids to prevent transplant rejection and combat graft-versus-host disease [8,12,21,49]. Besides these direct risks, patients who require radiotherapy or HSCT generally have severe ALL/LBL. Therefore, patients with ALL/LBL undergoing radiotherapy and HSCT should be closely monitored during follow-up. Finally, besides age and steroid use, L-ASP was a risk factor in the multivariate analysis (HR = 1.99 in all age groups and HR = 2.55 in the paediatric group), but not in the young adult group (Appendix A). Increasing evidence suggests that L-ASP is associated with an increased ON risk through indirect effects that influence glucocorticoid pharmacokinetics and cause hyperlipidaemia [7,26,33,50].

Previous studies have attempted to define factors that predispose patients to ON. In our multivariate analysis, age, steroid use and L-ASP were significant risk factors (*p* < 0.05). In our study and other studies, age is the most consistent and significant risk factor, with patients aged ≥ 10 years at the highest risk across treatment regimens and study groups in paediatric age [8,9,10,13,51]. Hormonal changes unique to adolescents may lead to increased ON susceptibility through different mechanisms, including increased local metabolic/perfusion requirements, skeletal maturation, coagulation system, or osseous blood vessel supply [27]. Corticosteroids are key for treating ALL/LBL and can cause adverse effects, including infection, hypertension, diabetes, neuropsychological disorders, and ON [52]. Larsen et al. reported that dexamethasone administered during induction benefited younger children but provided no benefit and was associated with higher ON risk among participants aged ≥ 10 years [24]. Both the steroid type and how steroids are used have influenced ON development [11,53]. In our study, the use of dexamethasone in the paediatric group was lower than that in the prednisone group; however, dexamethasone remains a risk factor for ON development. Similar to steroids, L-Asp is an important drug for treating ALL/LBL, and the intensive use of L-Asp leads to decreased dexamethasone clearance [54]. Steroids and L-ASP are key drugs in paediatric-inspired treatment protocols; therefore, further standard protocols should be considered to suggest the best combination to achieve complete remission of ALL/LBL and prevent ON. Various genetic risk factors for ON development and steroid-induced ON have been identified in large genome-wide studies over decades [7,10,18,55]. Our study is based on claims data; therefore, access to genetic information was unavailable. However, this is the first study related to the treatment of ALL/LBL in a nationwide Korean cohort. Considering the association of genetics and ethnicity with ON, this study will serve as an important basis for further evaluating ON management and prevention in paediatric and young adult patients with ALL/LBL in Koreans.

Our study had some limitations. The cumulative dose of steroids was an important risk factor in previous studies [7,22,24,27,52]; however, it was not available in our national claims data. Methotrexate is also a critical component of ALL/LBL therapy [24]; however, it was not evaluated in our study because of the lack of BMI to determine the total dose. We included ON based on diagnostic codes, but there was no information about its severity based on clinical symptoms, X-ray or MRI. There is no consistent treatment regimen in our study as it was based on national data, including all institutions in Korea. There was no information of ON patients without surgical treatment. Follow-up loss, medical treatment and improvement without treatment are all possible in these patients. Notably, voids in clinical and biological information are imposed by insurance-based data collection, making it impossible to generate risk estimates for ALL/LBL. However, most treatments of ALL/LBL are covered by national insurance; thus, it is valuable to estimate the incidence, clinical characteristics and risk factors in Korean children and young adult patients with ALL/LBL.

Despite its limitations, as this study was based on data from all paediatric, adolescent and young adult patients registered as ALL or LBL in Korea, it is expected to provide important insights into the consideration of ON after treatment of these patients.

## Figures and Tables

**Figure 1 jcm-11-02489-f001:**
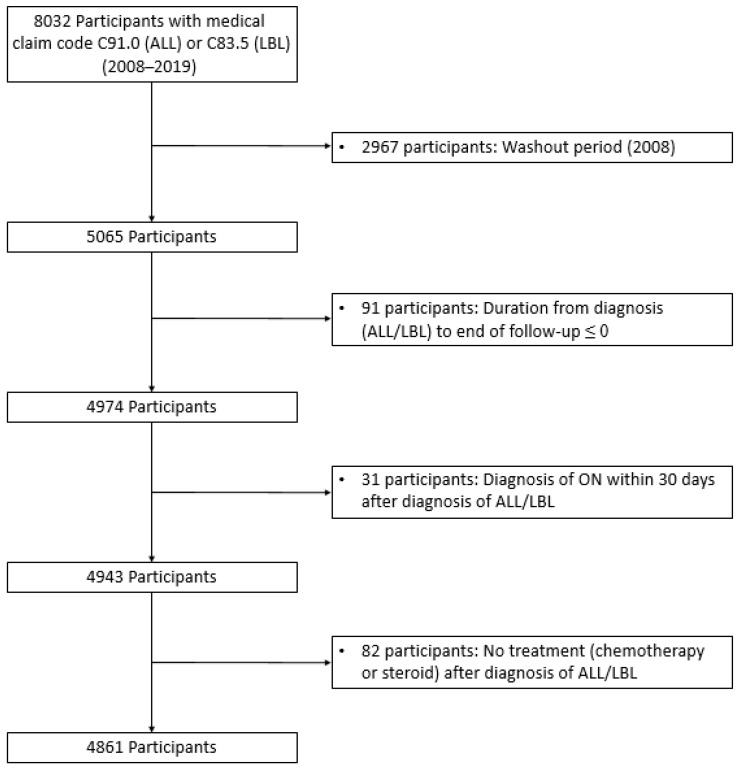
Participant enrolment flow diagram. Among 8056 participants, 4861 with ALL/LBL were included in the study. A total of 2967 patients were excluded owing to the washout period.

**Figure 2 jcm-11-02489-f002:**
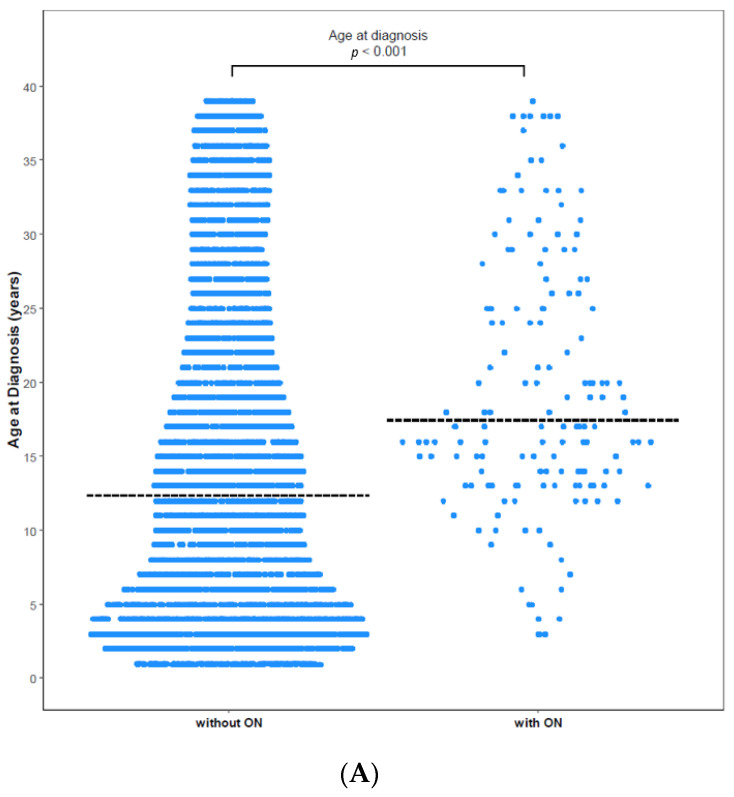
Age at diagnosis with and without ON among (**A**) total patients and (**B**) paediatric patients. (**A**) The mean age of patients without ON was 14.73 years, and that of patients with ON was 19.41 years (*p* < 0.0001). (**B**) The mean age of paediatric patients without ON was 8.20 years, and that of paediatric patients with ON was 13.28 years (*p* < 0.0001). Dashed horizontal lines reflect the median age at diagnosis of two groups.

**Figure 3 jcm-11-02489-f003:**
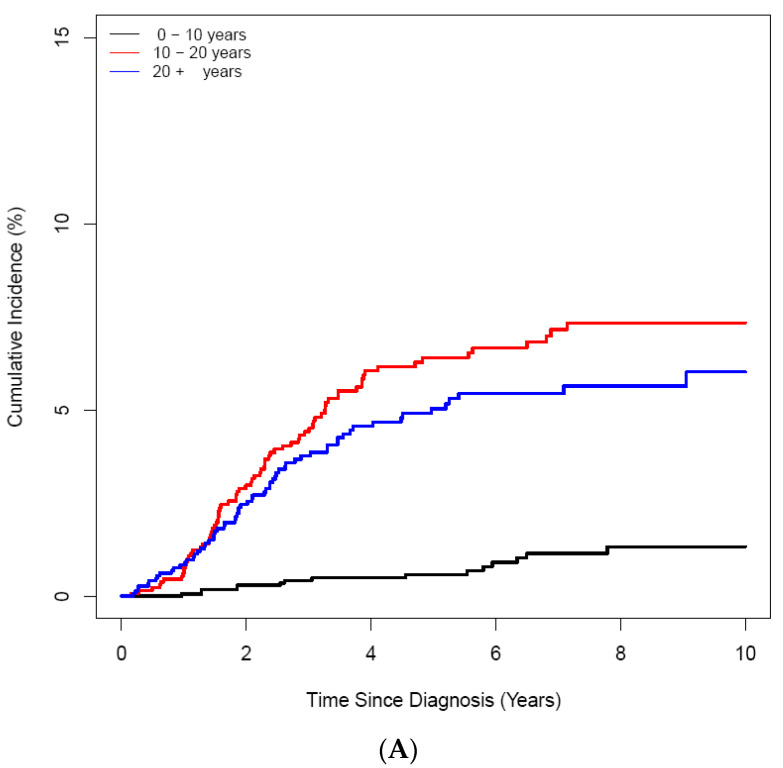
Cumulative incidence of ON according to age at the diagnosis of ALL/LBL in 4861 patients. (**A**) A 10-year scale, (**B**) 5-year scale (C) by age and sex. (**A**) During 10–20-years, the highest CI was noted, and during 15–20 years, patients had the highest CI (**B**). Girls showed higher CI than boys (**C**).

**Figure 4 jcm-11-02489-f004:**
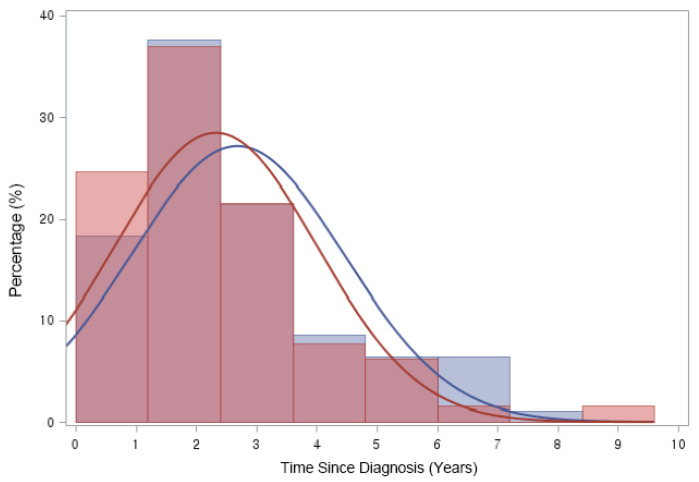
Time from ALL/LBL diagnosis to ON diagnosis. Time from cancer treatment to ON diagnosis was 2.54 (±1.74) years after ALL or LBL diagnosis in all patients. Approximately 90% of patients were diagnosed as having ON in 5 years after primary cancer. (Blue line: Paediatric group (2.69 ± 1.77 years), Red line: Adult group (2.33 ± 1.68 years)).

**Table 1 jcm-11-02489-t001:** Demographic characteristics of all the participants.

		All Participants	Paediatric Group	Adult Group	*p*-Value *
		(*n* = 4861)	(*n* = 3344)	(*n* = 1517)	
Risk Factors		Median (Q1, Q3) or *n* (%)	Median (Q1, Q3) or *n* (%)	Median (Q1, Q3) or *n* (%)	
Age		13	7	29	<0.001
		(5, 23)	(3, 13)	(24, 34)	
Age according to group	0 ≤ age < 5	1181	1181		
		(24.30)	(35.32)		
	5 ≤ age < 10	797	797		
		(16.40)	(23.83)		
	10 ≤ age < 15	719	719		
		(14.79)	(21.50)		
	15 ≤ age < 20	647	647		
		(13.31)	(19.35)		
	20 ≤ age	1517			
		(31.21)			
Sex	Male	2885	1973	912	0.462
		(59.35)	(59.00)	(60.12)	
Radiotherapy	Yes	995	474	521	<0.001
		(20.47)	(14.17)	(34.34)	
HSCT	Yes	1393	560	833	<0.001
		(28.66)	(16.75)	(54.91)	
Steroid use **	Dexamethasone	1111	544	567	<0.001
		(22.86)	(16.27)	(37.38)	
	Prednisone	2020	1666	354	
		(41.56)	(49.82)	(23.34)	
	Dexamethasone	1189	781	408	
	+Prednisone	(24.26)	(23.36)	(26.90)	
	Not used	541	353	188	
		(11.13)	(10.56)	(12.39)	
Asparaginase use	Yes	3318 (68.26)	2703 (81.83)	615	<0.001
				(40.54)	
Anthracycline use **	Yes	3149 (64.78)	1940 (58.01)	308	<0.001
				(20.30)	

* Wilcoxon rank sum test. HSCT, hematopoietic stem cell transplantation; ** during 1st 60 days after diagnosis.

**Table 2 jcm-11-02489-t002:** Risk factors of osteonecrosis.

Risk Factors		ALL			Paediatric Group (*n* = 3344)		
		No ON	ON	*p*-Value *	No ON	ON	*p*-Value *
		(*n* = 4703)	(*n* = 158)		(*n* = 3251)	(*n* = 93)	
		Median (Q1, Q3) or *n* (%)	Median (Q1, Q3) or *n* (%)		Median (Q1, Q3) or *n* (%)	Median (Q1, Q3) or *n* (%)	
Age		12 (4, 23)	17 (13, 26)	<0.001	7 (3, 13)	14 (12, 16)	<0.001
Age according to group	0 ≤ age < 5	1176 (99.58)	5	<0.001	1176 (99.58)	5	<0.001
			(0.42)			(0.42)	
	5 ≤ age < 10	787 (98.74)	10		787 (98.75)	10	
			(1.25)			(1.25)	
	10 ≤ age < 15	683 (94.99)	36		683 (94.99)	36	
			(5.01)			(5.01)	
	15 ≤ age < 20	605 (93.51)	42		605 (93.51)	42	
			(6.49)			(6.49)	
	20 ≤ age	1452 (95.72)	65		-	-	
			(4.28)				
Sex	Male	2800 (97.05)	85	0.149	1927 (97.67)	46	0.058
			(2.95)			(2.33)	
Radiotherapy	Yes	944	51	<0.001	447 (94.30)	27	<0.001
		−94.87	(5.13)			(5.7)	
HSCT	Yes	1329 (95.41)	64	<0.001	537 (95.89)	23	0.037
			(4.59)			(4.11)	
Steroid use **	Dexamethasone	1074 (96.67)	37	0.141	531 (97.61)	13	0.095
			(3.33)			(2.39)	
	Prednisone	1953 (96.68)	67		1619 (97.18)	47	
			(3.32)			(2.82)	
	Dexamethasone	1144 (96.22)	45		752 (96.29)	29	
	+Prednisone		(3.78)			(3.71)	
	Not used	532 (98.34)	9		349 (98.87)	4	
			(1.66)			(1.13)	
Asparaginase use	Yes	3205 (96.59)	113 (3.41)	0.371	2622 (97.00)	81	0.120
						(3.00)	
Anthracycline use **	Yes	3020 (95.90)	129 (4.10)	<0.001	1867 (96.24)	73 (3.76)	<0.001

* Wilcoxon rank sum test. HSCT, hematopoietic stem cell transplantation; ** during 1st 60 days after diagnosis.

**Table 3 jcm-11-02489-t003:** Univariate and multivariate analysis associated with AVN.

(1) All Patients								
Risk Factors			Univariate		Multivariate_ON		Multivariate_Death	
	Level	Median (Q1, Q3) or *n* (%)	HR	*p*-Value *	HR	*p*-Value *	HR	*p*-Value *
			(95% CI)		(95% CI)		(95% CI)	
Age		13 (5, 23)	1.047	<0.001	-		-	
			(1.034, 1.061)					
Age according to group	0 ≤ age < 5	1181	1 (ref)		1 (ref)		1 (ref)	
		(24.30)						
	5 ≤ age < 10	797	3.056	0.041	3.08	0.040	1.008	0.961
		(16.40)	(1.045, 8.935)		(1.053, 9.026)		(0.736, 1.380)	
	10 ≤ age < 15	719	12.508	<0.001	13.56	<0.001	1.520	0.005
		(14.79)	(4.911, 31.854)		(5.170, 35.545)		(1.135, 2.037)	
	15 ≤ age < 20	647	18.167	<0.001	23.35	<0.001	2.798	<0.001
		(13.31)	(7.191, 45.894)		(8.897, 61.261)		(2.125, 3.683)	
	20 ≤ age	1517	14.684	<0.001	19.88	<0.001	3.926	<0.001
		(31.21)	(5.915, 36.456)		(7.508, 52.639)		(3.026, 5.093)	
Sex	Male	2885	0.818	0.208	0.74	0.058	1.092	0.187
		(59.35)	(0.598, 1.119)		(0.539, 1.010)		(0.958, 1.244)	
Radiotherapy	Yes	1976	2.615	<0.001	1.31	0.173	1.567	<0.001
		(40.65)	(1.869, 3.659)		(0.887, 1.941)		(1.334, 1.841)	
HSCT	Yes	995	2.398	<0.001	1.001	0.100	1.664	<0.001
		(20.47)	(1.739, 3.307)		(0.669, 1.498)		(1.402, 1.976)	
Steroid use **	Dexamethasone	3866	2.623	0.001	2.85	0.012	1.690	0.001
		(79.53)	(1.264, 5.442)		(1.257, 6.463)		(1.238, 2.309)	
	Prednisone	1393	2.003	0.051	2.28	0.052	0.976	0.884
		(28.66)	(0.998, 4.018)		(0.991, 5.224)		(0.704, 1.353)	
	Dexamethasone	3468	2.736	0.006	2.73	0.020	1.394	0.046
	+Prednisone	(71.34)	(1.336, 5.602)		(1.169, 6.387)		(1.005, 1.934)	
	Not used	1111	1 (ref)		1 (ref)			
		(22.86)						
Asparaginase use	Yes	3318	1.101	0.584	1.99	<0.001	1.839	<0.001
		(68.26)	(0.779, 1.556)		(1.302, 3.061)		(1.569, 2.154)	
Anthracycline use **	Yes	3149	2.763	<0.001	0.93	0.776	1.306	0.010
		(64.78)	(1.846, 4.135)		(0.563, 1.536)			
**(2) Paediatric Age Group**								
**Risk Factors**			**Univariate**		**Multivariate_ON**		**Multivariate_Death**	
	**Level**	**Median (Q1, Q3) or** ***n* (%)**	**HR**	** *p* ** **-Value ***	**HR**	** *p* ** **-Value ***	**HR**	** *p* ** **-Value ***
			**(95% CI)**		**(95% CI)**		**(95% CI)**	
Age		7	1.189	<0.001				
		(3, 13)	(1.14, 1.24)					
Age according to group	0 ≤ age < 5	1181	1 (ref)		1 (ref)		1 (ref)	
		(35.32)						
	5 ≤ age < 10	797	3.070	0.041	3.079	0.040	0.889	0.467
		(23.83)	(1.05, 8.98)		(1.05, 9.03)		(0.647, 1.222)	
	10 ≤ age < 15	719	12.543	<0.001	14.567	<0.001	1.048	0.769
		(21.50)	(4.92, 31.96)		(5.42, 39.17)		(0.765, 1.437)	
	15 ≤ age < 20	647	18.274	<0.001	26.757	<0.001	1.787	<0.001
		(19.35)	(7.23, 46.20)		(9.87, 72.51)		(1.315, 2.428)	
Sex	Male	1973	0.683	0.067	0.582	0.009	1.059	0.569
		(59.00)	(0.46, 1.03)		(0.39, 0.88)		(0.869, 1.291)	
Radiotherapy	Yes	474	3.580	<0.001	1.686	0.052	2.333	<0.001
		(14.17)	(2.28, 5.61)		(0.99, 2.86)		(1.785, 3.049)	
HSCT	Yes	560	2.323	0.001	0.848	0.562	2.493	<0.001
		(16.75)	(1.45, 3.73)		(0.49, 1.48)		(1.908, 3.258)	
Steroid use **	Dexamethasone	544	2.291	0.147	3.571	0.040	1.530	0.094
		(16.27)	(0.75, 7.03)		(1.06, 12.00)		(0.930, 2.518)	
	Prednisone	1666	2.327	0.105	2.748	0.107	0.794	0.384
		(49.82)	(0.84, 6.46)		(0.80, 9.40)		(0.473, 1.334)	
	Dexamethasone	781	3.328	0.024	3.529	0.050	1.217	0.464
	+Prednisone	(23.36)	(1.17, 9.47)		(1.00,12.42)		(0.720, 2.056)	
	Not used	353	1 (ref)		1 (ref)			
		(10.56)						
Asparaginase use	Yes	2703	1.611	0.123	2.547	0.012	1.738	<0.001
		(81.83)	(0.88, 2.96)		(1.23, 5.28)		(1.297, 2.330)	
Anthracycline use **	Yes	1940	2.642	<0.001	0.755	0.377	1.571	0.001
		(58.01)	(1.61, 4.33)		(0.41, 1.41		(1.195, 2.064)	

* Wilcoxon rank sum test. HSCT, hematopoietic stem cell transplantation; ** during 1st 60 days after diagnosis; HR, hazard ratio.

## Data Availability

The data were provided in the manuscript.

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
