# Peer review of "Osteonecrosis in Korean Paediatric and Young Adults with Acute Lymphoblastic Leukaemia or Lymphoblastic Lymphoma: A Nationwide Epidemiological Study"

_jcm, 2022, doi:10.3390/jcm11092489_

Round 1
Reviewer 1 Report
In the presented paper the incidence and associated risk factors of osteonecrosis after acute leukemia/lymphoma treatment. The study is overall quite well conducted and written, and it is notably thorough and covers wide cohort. This is also great that authors do realize their limitations, especially in the first rising question on the prevalence of certain stages and grades of osteonecrosis. However, I have some questions and remarks:
1. Applied drugs are analyses, however, no exact treatment protocols were mentioned. Were there any prevalent protocols and significant differences between used protocols/within participating institutions in osteonecrosis rates?
2. ALL/LBL is wide and heterogeneous group of diseases. Was there any evidence of a correlation between osteonecrosis prevalence and type/genetics of primary leukemia/lymphoma?
3. ALL/LBL is life-threatening disease as well. What was the overall survival in the described cohort? It would be rather informative to compare overall survival and osteonecrosis-free survival.
4. I wonder why 20 y.o. was taken as a threshold between pediatric and adult cohorts, not 18 or 21.
5. The authors did thorough statistical analysis, however, they should stick to median rather than mean values when describing age of compared groups. Exact p values should be stated. Figure 3 also looks quite blind without actual numbers.
6. Small mistakes include extra - in line 31. The manuscripts could also be shortened.
Author Response
Reviewer 1
- Applied drugs are analyses, however, no exact treatment protocols were mentioned. Were there any prevalent protocols and significant differences between used protocols/within participating --institutions in osteonecrosis rates?
; Thank you for your comment. It would be very nice if we had included the exact treatment protocols of the patients. However, because of the limitation of data from the Korean Health Insurance and Review Assessment (HIRA), it is hard to differentiate exact treatment protocols in each person (We are not able to know the exact schedules and subscription doses of the drug). Moreover, this study includes all patients in South Korea who were diagnosed as ALL/LBL and has been treated under the coverage of the national insurance system in South Korea. It is not based on the multicenter study, but on the national cohort, therefore, we did not contact all the institutions for their treatment protocols. It is hard to know and compare the differences between certain protocols, however we tried to find the trends of the drug used while focusing on the incidence of ON. Even though this is not prospective study based on the well-structured protocol, our research is based on nationwide epidemiology study.
- ALL/LBL is wide and heterogeneous group of diseases. Was there any evidence of a correlation between osteonecrosis prevalence and type/genetics of primary leukemia/lymphoma?
; Thank you for your comment, the Korean HIRA database does not include clinical laboratory results or results of the bone marrow study. We used ICD codes to define ALL/LBL, however it was impossible to differentiate genetic types of ALL/LBL in this study. But as reviewer’s comment, one of assumed factor related to osteonecrosis in paediatric and young adults with acute lymphoblastic leukaemia or lymphoblastic lymphoma is ethnicity. We have national cohort data (covers approximately 98% of the total population) based on relative mono-ethnicity.
ALL/LBL is life-threatening disease as well. What was the overall survival in the described cohort? It would be rather informative to compare overall survival and osteonecrosis-free survival.
; We highly agree the reviewer’s comment. For the overall survival (OS), we should access data from the Korea National Statistics Office (KNSO), not HIRA. Because there are limitations to find out the exact expire date from the HIRA database, we did not include accurate OS in this study. However during data analysis, we considered survival data from death date. We further analyze our data and modified table with death data as reviewer’s comment.
Table 3
- All patients
|
Risk factors |
  |
  |
Univariate |
Multivariate_ON |
Multivariate_Death |
|||
|
  |
Level |
Median (Q1, Q3) or |
HR |
p-value* |
HR |
p-value* |
HR |
p-value* |
|
(95% CI) |
(95% CI) |
(95% CI) |
||||||
|
Age |
  |
13 (5, 23) |
1.047 |
<0.001 |
- |
  |
- |
  |
|
(1.034, 1.061) |
||||||||
|
Age according to group |
0≤age<5 |
1181 |
1 (ref) |
  |
1 (ref) |
  |
1 (ref) |
  |
|
(24.30) |
||||||||
|
5≤age<10 |
797 |
3.056 |
0.041 |
3.08 |
0.040 |
1.008 |
0.961 |
|
|
(16.40) |
(1.045, 8.935) |
(1.053, 9.026) |
(0.736, 1.380) |
|||||
|
10≤age<15 |
719 |
12.508 |
<0.001 |
13.56 |
<0.001 |
1.520 |
0.005 |
|
|
(14.79) |
(4.911, 31.854) |
(5.170, 35.545) |
(1.135, 2.037) |
|||||
|
15≤age<20 |
647 |
18.167 |
<0.001 |
23.35 |
<0.001 |
2.798 |
<0.001 |
|
|
(13.31) |
(7.191, 45.894) |
(8.897, 61.261) |
(2.125, 3.683) |
|||||
|
20≤age |
1517 |
14.684 |
<0.001 |
19.88 |
<0.001 |
3.926 |
<0.001 |
|
|
(31.21) |
(5.915, 36.456) |
(7.508, 52.639) |
(3.026, 5.093) |
|||||
|
Sex |
Male |
2885 |
0.818 |
0.208 |
0.74 |
0.058 |
1.092 |
0.187 |
|
(59.35) |
(0.598, 1.119) |
(0.539, 1.010) |
(0.958, 1.244) |
|||||
|
Radiotherapy |
Yes |
1976 |
2.615 |
<0.001 |
1.31 |
0.173 |
1.567 |
<0.001 |
|
(40.65) |
(1.869, 3.659) |
(0.887, 1.941) |
(1.334, 1.841) |
|||||
|
HSCT |
Yes |
995 |
2.398 |
<0.001 |
1.001 |
0.100 |
1.664 |
<0.001 |
|
(20.47) |
(1.739, 3.307) |
(0.669, 1.498) |
(1.402, 1.976) |
|||||
|
Steroid use** |
Dexamethasone |
3866 |
2.623 |
0.001 |
2.85 |
0.012 |
1.690 |
0.001 |
|
(79.53) |
(1.264, 5.442) |
(1.257, 6.463) |
(1.238, 2.309) |
|||||
|
Prednisone |
1393 |
2.003 |
0.051 |
2.28 |
0.052 |
0.976 |
0.884 |
|
|
(28.66) |
(0.998, 4.018) |
(0.991, 5.224) |
(0.704, 1.353) |
|||||
|
Dexamethasone |
3468 |
2.736 |
0.006 |
2.73 |
0.020 |
1.394 |
0.046 |
|
|
+ Prednisone |
(71.34) |
(1.336, 5.602) |
(1.169, 6.387) |
(1.005, 1.934) |
||||
|
Not used |
1111 |
1 (ref) |
  |
1 (ref) |
  |
  |
  |
|
|
(22.86) |
||||||||
|
Asparaginase use |
Yes |
3318 |
1.101 |
0.584 |
1.99 |
<0.001 |
1.839 |
<0.001 |
|
(68.26) |
(0.779, 1.556) |
(1.302, 3.061) |
(1.569, 2.154) |
|||||
|
Anthracycline use** |
Yes |
3149 |
2.763 |
<0.001 |
0.93 |
0.776 |
1.306 |
0.010 |
|
(64.78) |
(1.846, 4.135) |
(0.563, 1.536) |
  |
|||||
|
*Wilcoxon rank sum test |
||||||||
|
HSCT, hematopoietic stem cell transplantation; **during 1st 60 days after diagnosis |
||||||||
- Pediatric patients
|
Risk factors |
  |
  |
Univariate |
Multivariate_ON |
Multivariate_Death |
|||
|
  |
Level |
Median (Q1, Q3) or |
HR |
p-value* |
HR |
p-value* |
HR |
p-value* |
|
(95% CI) |
(95% CI) |
(95% CI) |
||||||
|
Age |
  |
7 |
1.189 |
<0.001 |
  |
  |
  |
  |
|
(3, 13) |
(1.14, 1.24) |
|||||||
|
Age according to group |
0≤age<5 |
1181 |
1 (ref) |
  |
1 (ref) |
  |
1 (ref) |
  |
|
(35.32) |
||||||||
|
5≤age<10 |
797 |
3.070 |
0.041 |
3.079 |
0.040 |
0.889 |
0.467 |
|
|
(23.83) |
(1.05, 8.98) |
(1.05, 9.03) |
(0.647, 1.222) |
|||||
|
10≤age<15 |
719 |
12.543 |
<0.001 |
14.567 |
<0.001 |
1.048 |
0.769 |
|
|
(21.50) |
(4.92, 31.96) |
(5.42, 39.17) |
(0.765, 1.437) |
|||||
|
15≤age<20 |
647 |
18.274 |
<0.001 |
26.757 |
<0.001 |
1.787 |
<0.001 |
|
|
(19.35) |
(7.23, 46.20) |
(9.87, 72.51) |
(1.315, 2.428) |
|||||
|
Sex |
Male |
1973 |
0.683 |
0.067 |
0.582 |
0.009 |
1.059 |
0.569 |
|
(59.00) |
(0.46, 1.03) |
(0.39, 0.88) |
(0.869, 1.291) |
|||||
|
Radiotherapy |
Yes |
474 |
3.580 |
<0.001 |
1.686 |
0.052 |
2.333 |
<0.001 |
|
(14.17) |
(2.28, 5.61) |
(0.99, 2.86) |
(1.785, 3.049) |
|||||
|
HSCT |
Yes |
560 |
2.323 |
0.001 |
0.848 |
0.562 |
2.493 |
<0.001 |
|
(16.75) |
(1.45, 3.73) |
(0.49, 1.48) |
(1.908, 3.258) |
|||||
|
Steroid use** |
Dexamethasone |
544 |
2.291 |
0.147 |
3.571 |
0.040 |
1.530 |
0.094 |
|
(16.27) |
(0.75, 7.03) |
(1.06, 12.00) |
(0.930, 2.518) |
|||||
|
Prednisone |
1666 |
2.327 |
0.105 |
2.748 |
0.107 |
0.794 |
0.384 |
|
|
(49.82) |
(0.84, 6.46) |
(0.80, 9.40) |
(0.473, 1.334) |
|||||
|
Dexamethasone |
781 |
3.328 |
0.024 |
3.529 |
0.050 |
1.217 |
0.464 |
|
|
+ Prednisone |
(23.36) |
(1.17, 9.47) |
(1.00,12.42) |
(0.720, 2.056) |
||||
|
Not used |
353 |
1 (ref) |
  |
1 (ref) |
  |
  |
  |
|
|
(10.56) |
||||||||
|
Asparaginase use |
Yes |
2703 |
1.611 |
0.123 |
2.547 |
0.012 |
1.738 |
<0.001 |
|
(81.83) |
(0.88, 2.96) |
(1.23, 5.28) |
(1.297, 2.330) |
|||||
|
Anthracycline use** |
Yes |
1940 |
2.642 |
<0.001 |
0.755 |
0.377 |
1.571 |
0.001 |
|
(58.01) |
(1.61, 4.33) |
(0.41, 1.41 |
(1.195, 2.064) |
|||||
|
*Wilcoxon rank sum test |
||||||||
|
HSCT, hematopoietic stem cell transplantation; **during 1st 60 days after diagnosis |
||||||||
I wonder why 20 y.o. was taken as a threshold between pediatric and adult cohorts, not 18 or 21.
; Thank you for your question. Age definition of children and AYA age group varies by nation or by protocols. In our country, age over 20 years old are considered as legally adult. Also, we first set the age under 40 years old (which is the upper limit age of AYA group) to be included in the study, and then decided to further differentiate pediatric group into 4 groups with 5 years-interval.
- The authors did thorough statistical analysis, however, they should stick to median rather than mean values when describing age of compared groups. Exact p values should be stated. Figure 3 also looks quite blind without actual numbers.
; Thank you for your comments. We modified statistical analysis in the manuscript and tables accordingly.
Statistical analysis
Comparisons between the two groups were conducted using Wilcoxon rank sum test for continuous variables and chi-square tests for categorical variables.
|
  |
  |
All participants (n=4861) |
Pediatric group (n=3344) |
Adult group (n=1517) |
p-value* |
|
Risk factors |
Median (Q1, Q3) or n (%) |
|
|
||
|
Age |
  |
13 (5, 23) |
7 (3, 13) |
29 (24, 34) |
<0.01 |
* Wilcoxon rank sum test
Table 2
|
Risk factors |
ALL |
Pediatric group (n=3344) |
|||||
|
No AVN (n=4703) |
AVN (n=158) |
p-value |
No AVN (n=3251) |
AVN (n=93) |
p-value* |
||
|
Age |
12 (4, 23) |
17 (13, 26) |
<0.01 |
7 (3, 13) |
14 (12, 16) |
<0.01 |
|
* Wilcoxon rank sum test
Table 3. Univariate and multivariate analysis associated with AVN
- All patients
|
Risk factors |
  |
  |
Univariate |
Multivariate |
||
|
Median (Q1, Q3) or n (%) |
HR (95% CI) |
p-value |
HR (95% CI) |
p-value |
||
|
Age |
13 (5, 23) |
1.04 (1.03, 1.05) |
<0.01 |
- |
||
- Pediatric age group
|
Risk factors |
|
|
Univariate |
Multivariate |
||
|
Level |
Median (Q1, Q3) or n (%) |
HR (95% CI) |
p-value |
HR (95% CI) |
p-value |
|
|
Age |
7 (3, 13) |
1.19 (1.14, 1.24) |
<0.01 |
|||
- Small mistakes include extra – in line 31. The manuscripts could also be shortened.
; We apologize our typos and thank you so much for your meticulous review. Also, we shortened our manuscripts as the reviewer’s comment. As other reviewer’s comments, we removed line 268-274 and line 293-295 (Not critically related to our results)

Reviewer 2 Report
It is well designed and interesting study concerning osteonecrosis in Korean paediatric and young adults with acute lymphoblastic leukaemia or lymphoblastic lymphoma. The authors used national health insurance claims data to assess cumulative incidence of osteonecrosis and to identify risk factors related to osteonecrosis development after diagnosing of acute lymphoblastic leukaemia or lymphoblastic lymphoma.
The study is well written, however some minor corrections are needed as listed below:
- Abbreviation used in the tables or figures should be explained
- Table 1 and 3: Std and % should be in brackets
- Line 149:” The mean age at diagnosis” – it should be “at ALL/LBL diagnosis”
- Figures – too small font size on the graphs, it is illegible
- Figure 4 – it should be explained what the blue and red colour means
- Table 2 and Table 1 in supplement : It should be explained what the numbers in brackets or after dashes mean.
Authors reported that steroid and asparaginase use were factors associated with ON and concluded that patients with ALL/LBL who used steroid and asparaginase should be closely monitored during follow-up. As steroids and asparaginase are basic regimens used in ALL/LBL in children it is difficult to imagine the group of patients cured from ALL/LBL with no steroids and no asparaginase administration. It should be explained.
Author Response
Reviewer 2
The study is well written, however some minor corrections are needed as listed below:
- Abbreviation used in the tables or figures should be explained
- Table 1 and 3: Std and % should be in brackets
- Line 149:” The mean age at diagnosis” – it should be “at ALL/LBL diagnosis”
- Figures – too small font size on the graphs, it is illegible
- Figure 4 – it should be explained what the blue and red colour means
- Table 2 and Table 1 in supplement: It should be explained what the numbers in brackets or after dashes mean.
; Thank you for your comments. We modified our manuscript as reviewer’s comments.
Authors reported that steroid and asparaginase use were factors associated with ON and concluded that patients with ALL/LBL who used steroid and asparaginase should be closely monitored during follow-up. As steroids and asparaginase are basic regimens used in ALL/LBL in children it is difficult to imagine the group of patients cured from ALL/LBL with no steroids and no asparaginase administration. It should be explained.
; We appreciate the reviewer’s comments. We totally agree that steroids and asparaginase are key drugs in treating ALL/LBL. However, there are many different protocols using different steroids, doses of steroids and asparaginases. Moreover, in South Korea there is still no national guideline of the ALL/LBL treatment: although “Korean Society of Pediatric Hematology and Oncology” is working on set up a unified protocol guideline. Hence, we were not suggesting that we should omit steroid or asparaginase during the ALL/LBL treatment. What we emphasize in Line 268-271; Steroids and L-ASP are key drugs in paediatric-inspired treatment protocols; therefore, further standard protocols should be considered to suggest the best combination to achieve complete remission of ALL/LBL and prevent ON. This is what we need to put some effort to make attentive protocols to prevent ON as we can.

Reviewer 3 Report
It is an interesting manuscript and first data about ON and its risk factors in a large cohort of Korean patients with ALL including children and young adults. The data are unbiased since they are retrieved from the National insurance claims database confirming previously published data.
As it was mentioned in the manuscript there are some data missing from the claims database, e.g. location, severity and treatment of ON, which should be the base for the data! The treatment period of corticoids and anthracylines is restricted to the early treatment phase, not the whole therapy time, which could be skewing the data, since the amount of corticoids might have an impact on ON. The treatment protocols/modalities were not mentioned for the patients. Do different protocols (pediatric inspired vs adult protocols) have an influence on ON? Regarding therapy it is not quite clear, what radiotherapy include: does it refer to TBI as part of the conditioning regimen prior to HSCT or does it include all irradiation performed in the patients? It seems that different age groups in children have an impact on ON, since the authors have divided them in age categories, whereas patients >20 years were not subcategorized. Are there not any differences in different age groups in this patient group? Overall it is sometimes confusing to follow the different groups (all patients, pediatric patients, young adults). Not all groups are mentioned consistently, e.g. as comparison. Is there a rationale behind it?
Since the data do not seem to be normally distributed (e.g. figure 2), median/range instead of mean/SD and their respective statistical tests should be used. CI should not be investigated with the Kaplan Meier method, but with the Gray Fine method.
Specific topics:
- Page 3 line 116: why date of ON as end of follow up? No follow up after ON?
- Tables 1,3: what do the numbers in the column mean?
- Table 2: what do the numbers in the column mean? Different than table 1
- Figure 2: division of patients >20 years in age categories?
- Figure 4: what is the meaning of this figure?
- Page 9 lines 237-238: The subset of patients receiving these treatments is exposed to total-body radiation and considerably higher doses of corticosteroids to prevent transplant rejection and combat graft-versus-host disease – is that an assumption or based on patients´data?
- Page 9 line 242: Finally, besides age and steroid use, L-ASP was a risk factor in the multivariate analysis (HR=1.99 in all age groups and HR=2.55 in the paediatric group) –significance only in the whole group or pediatric group influencing the whole group? What about young adults?
- Page 9 lines 245-252: What is the meaning of this sentence regarding ON?
- Page 9 lines 255-256: In our study and other studies, age is the most consistent and significant risk 255 factor, with patients aged ≥10 years at the highest risk across treatment regimens and 256 study groups [8-10,13,53]. – what are the data in young adults? Reference here are pediatric (adolescent) data.
- Page 9 lines 265-266: in our study, the use of dexamethasone in the paediatric group was lower than that in the prednisone group – what is the meaning of this sentence? Why highlight the difference between dexamethasone and prednisone?
- Page 9 lines 271-272: ON occurs more frequently in white patients than in black or Hispanic patients. – What is the purpose of this sentence, when describing Korean patients?
Author Response
Reviewer 3
- As it was mentioned in the manuscript there are some data missing from the claims database, e.g. location, severity and treatment of ON, which should be the base for the data!
; We appreciate reviewer’s comment. Due to limitation of the national cohort data, it is prohibited to access patients’ chart as personal data. But we had data of the location of ON and added as the reviewer’s comment and added on line 224. The treatment of ON was described in line 201 (new line 203).
- The treatment period of corticoids and Anthracyclines is restricted to the early treatment phase, not the whole therapy time, which could be skewing the data, since the amount of corticoids might have an impact on ON. The treatment protocols/modalities were not mentioned for the patients. Do different protocols (pediatric inspired vs adult protocols) have an influence on ON?
; Thank you for your kind, thoughtful comments. About the whole drug usage and protocols/modalities of the treatment, we agree that if those data were added it would be much informative and objective. However, it is hard to differentiate exact treatment protocols in each person (we are not able to know the exact day-to-day schedules and subscription doses of the drugs) from the Korean Health Insurance and Review Assessment (HIRA) data. That is why we couldn’t calculate the cumulative doses of drugs and differentiate patients by protocol. We were only able to know which drugs are used in certain period. We know most pediatric (inspired) regimens use asparaginase, and during induction period, mostly high-risk group children receive anthracycline (most standard risk don’t), so we tried to differentiate patients’ group according to the drug usage, more focusing on the incidence of ON.
- Regarding therapy it is not quite clear, what radiotherapy include: does it refer to TBI as part of the conditioning regimen prior to HSCT or does it include all irradiation performed in the patients?
; Thank you for your question. Radiotherapy included TBI and any brain, spinal or local radiotherapy ALL/LBL patients would possibly receive.
- It seems that different age groups in children have an impact on ON, since the authors have divided them in age categories, whereas patients >20 years were not subcategorized. Are there not any differences in different age groups in this patient group? Overall it is sometimes confusing to follow the different groups (all patients, pediatric patients, young adults). Not all groups are mentioned consistently, e.g. as comparison. Is there a rationale behind it?
; Thank you for your question. Out study aim is to find treatment related ON in pediatric and AYA age group. Hence >20 years old~39 years old age was considered as one group which is defined as AYA group among adults. We would consider to check age impact among adults in further studies.
- Since the data do not seem to be normally distributed (e.g. figure 2), median/range instead of mean/SD and their respective statistical tests should be used. CI should not be investigated with the Kaplan Meier method, but with the Gray Fine method.
; We appreciate your valuable comments. We reviewed statistical analyses what we had conducted in the primary analyses and found that we definitely used Gray’s method in cumulative incidence calculation after considering the competing events. However, we described the statistical method incorrectly in the manuscript. We modified the manuscript as follows:
Statistical analysis
Line 117
The Gray’s method was used to estimate the cumulative incidence (Cuml) of ON according to the age group.
We also modified statistical analysis in the manuscript and tables accordingly (median/range).
Line 114
Comparisons between the two groups were conducted using Wilcoxon rank sum test for continuous variables and chi-square tests for categorical variables.
|
  |
  |
All participants (n=4861) |
Pediatric group (n=3344) |
Adult group (n=1517) |
p-value* |
|
Risk factors |
median (Q1, Q3) or n (%) |
|
|
||
|
Age |
  |
13 (5, 23) |
7 (3, 13) |
29 (24, 34) |
<0.01 |
* Wilcoxon rank sum test
Table 2
|
Risk factors |
ALL |
Pediatric group (n=3344) |
|||||
|
No AVN (n=4703) |
AVN (n=158) |
p-value |
No AVN (n=3251) |
AVN (n=93) |
p-value* |
||
|
Age |
12 (4, 23) |
17 (13, 26) |
<0.01 |
7 (3, 13) |
14 (12, 16) |
<0.01 |
|
* Wilcoxon rank sum test
Table 3. Univariate and multivariate analysis associated with AVN
- All patients
|
Risk factors |
  |
  |
Univariate |
Multivariate |
||
|
median (Q1, Q3) or n (%) |
HR (95% CI) |
p-value |
HR (95% CI) |
p-value |
||
|
Age |
13 (5, 23) |
1.04 (1.03, 1.05) |
<0.01 |
- |
||
- Pediatric age group
|
Risk factors |
|
|
Univariate |
Multivariate |
||
|
Level |
median (Q1, Q3) or n (%) |
HR (95% CI) |
p-value |
HR (95% CI) |
p-value |
|
|
Age |
7 (3, 13) |
1.19 (1.14, 1.24) |
<0.01 |
|||
Specific topics:
- Page 3 line 116: why date of ON as end of follow up? No follow up after ON?
; Thank you for your question. Our study is related to risk of the developing ON, so date of ON was one of primary end point. But we also analyzed death so we further analyze the data and modified tables.
Table 3
- All patients
|
Risk factors |
  |
  |
Univariate |
Multivariate_ON |
Multivariate_Death |
|||
|
  |
Level |
Median (Q1, Q3) or |
HR |
p-value* |
HR |
p-value* |
HR |
p-value* |
|
(95% CI) |
(95% CI) |
(95% CI) |
||||||
|
Age |
  |
13 (5, 23) |
1.047 |
<0.001 |
- |
  |
- |
  |
|
(1.034, 1.061) |
||||||||
|
Age according to group |
0≤age<5 |
1181 |
1 (ref) |
  |
1 (ref) |
  |
1 (ref) |
  |
|
(24.30) |
||||||||
|
5≤age<10 |
797 |
3.056 |
0.041 |
3.08 |
0.040 |
1.008 |
0.961 |
|
|
(16.40) |
(1.045, 8.935) |
(1.053, 9.026) |
(0.736, 1.380) |
|||||
|
10≤age<15 |
719 |
12.508 |
<0.001 |
13.56 |
<0.001 |
1.520 |
0.005 |
|
|
(14.79) |
(4.911, 31.854) |
(5.170, 35.545) |
(1.135, 2.037) |
|||||
|
15≤age<20 |
647 |
18.167 |
<0.001 |
23.35 |
<0.001 |
2.798 |
<0.001 |
|
|
(13.31) |
(7.191, 45.894) |
(8.897, 61.261) |
(2.125, 3.683) |
|||||
|
20≤age |
1517 |
14.684 |
<0.001 |
19.88 |
<0.001 |
3.926 |
<0.001 |
|
|
(31.21) |
(5.915, 36.456) |
(7.508, 52.639) |
(3.026, 5.093) |
|||||
|
Sex |
Male |
2885 |
0.818 |
0.208 |
0.74 |
0.058 |
1.092 |
0.187 |
|
(59.35) |
(0.598, 1.119) |
(0.539, 1.010) |
(0.958, 1.244) |
|||||
|
Radiotherapy |
Yes |
1976 |
2.615 |
<0.001 |
1.31 |
0.173 |
1.567 |
<0.001 |
|
(40.65) |
(1.869, 3.659) |
(0.887, 1.941) |
(1.334, 1.841) |
|||||
|
HSCT |
Yes |
995 |
2.398 |
<0.001 |
1.001 |
0.100 |
1.664 |
<0.001 |
|
(20.47) |
(1.739, 3.307) |
(0.669, 1.498) |
(1.402, 1.976) |
|||||
|
Steroid use** |
Dexamethasone |
3866 |
2.623 |
0.001 |
2.85 |
0.012 |
1.690 |
0.001 |
|
(79.53) |
(1.264, 5.442) |
(1.257, 6.463) |
(1.238, 2.309) |
|||||
|
Prednisone |
1393 |
2.003 |
0.051 |
2.28 |
0.052 |
0.976 |
0.884 |
|
|
(28.66) |
(0.998, 4.018) |
(0.991, 5.224) |
(0.704, 1.353) |
|||||
|
Dexamethasone |
3468 |
2.736 |
0.006 |
2.73 |
0.020 |
1.394 |
0.046 |
|
|
+ Prednisone |
(71.34) |
(1.336, 5.602) |
(1.169, 6.387) |
(1.005, 1.934) |
||||
|
Not used |
1111 |
1 (ref) |
  |
1 (ref) |
  |
  |
  |
|
|
(22.86) |
||||||||
|
Asparaginase use |
Yes |
3318 |
1.101 |
0.584 |
1.99 |
<0.001 |
1.839 |
<0.001 |
|
(68.26) |
(0.779, 1.556) |
(1.302, 3.061) |
(1.569, 2.154) |
|||||
|
Anthracycline use** |
Yes |
3149 |
2.763 |
<0.001 |
0.93 |
0.776 |
1.306 |
0.010 |
|
(64.78) |
(1.846, 4.135) |
(0.563, 1.536) |
  |
|||||
|
*Wilcoxon rank sum test |
||||||||
|
HSCT, hematopoietic stem cell transplantation; **during 1st 60 days after diagnosis |
||||||||
- Pediatric patients
|
Risk factors |
  |
  |
Univariate |
Multivariate_ON |
Multivariate_Death |
|||
|
  |
Level |
Median (Q1, Q3) or |
HR |
p-value* |
HR |
p-value* |
HR |
p-value* |
|
(95% CI) |
(95% CI) |
(95% CI) |
||||||
|
Age |
  |
7 |
1.189 |
<0.001 |
  |
  |
  |
  |
|
(3, 13) |
(1.14, 1.24) |
|||||||
|
Age according to group |
0≤age<5 |
1181 |
1 (ref) |
  |
1 (ref) |
  |
1 (ref) |
  |
|
(35.32) |
||||||||
|
5≤age<10 |
797 |
3.070 |
0.041 |
3.079 |
0.040 |
0.889 |
0.467 |
|
|
(23.83) |
(1.05, 8.98) |
(1.05, 9.03) |
(0.647, 1.222) |
|||||
|
10≤age<15 |
719 |
12.543 |
<0.001 |
14.567 |
<0.001 |
1.048 |
0.769 |
|
|
(21.50) |
(4.92, 31.96) |
(5.42, 39.17) |
(0.765, 1.437) |
|||||
|
15≤age<20 |
647 |
18.274 |
<0.001 |
26.757 |
<0.001 |
1.787 |
<0.001 |
|
|
(19.35) |
(7.23, 46.20) |
(9.87, 72.51) |
(1.315, 2.428) |
|||||
|
Sex |
Male |
1973 |
0.683 |
0.067 |
0.582 |
0.009 |
1.059 |
0.569 |
|
(59.00) |
(0.46, 1.03) |
(0.39, 0.88) |
(0.869, 1.291) |
|||||
|
Radiotherapy |
Yes |
474 |
3.580 |
<0.001 |
1.686 |
0.052 |
2.333 |
<0.001 |
|
(14.17) |
(2.28, 5.61) |
(0.99, 2.86) |
(1.785, 3.049) |
|||||
|
HSCT |
Yes |
560 |
2.323 |
0.001 |
0.848 |
0.562 |
2.493 |
<0.001 |
|
(16.75) |
(1.45, 3.73) |
(0.49, 1.48) |
(1.908, 3.258) |
|||||
|
Steroid use** |
Dexamethasone |
544 |
2.291 |
0.147 |
3.571 |
0.040 |
1.530 |
0.094 |
|
(16.27) |
(0.75, 7.03) |
(1.06, 12.00) |
(0.930, 2.518) |
|||||
|
Prednisone |
1666 |
2.327 |
0.105 |
2.748 |
0.107 |
0.794 |
0.384 |
|
|
(49.82) |
(0.84, 6.46) |
(0.80, 9.40) |
(0.473, 1.334) |
|||||
|
Dexamethasone |
781 |
3.328 |
0.024 |
3.529 |
0.050 |
1.217 |
0.464 |
|
|
+ Prednisone |
(23.36) |
(1.17, 9.47) |
(1.00,12.42) |
(0.720, 2.056) |
||||
|
Not used |
353 |
1 (ref) |
  |
1 (ref) |
  |
  |
  |
|
|
(10.56) |
||||||||
|
Asparaginase use |
Yes |
2703 |
1.611 |
0.123 |
2.547 |
0.012 |
1.738 |
<0.001 |
|
(81.83) |
(0.88, 2.96) |
(1.23, 5.28) |
(1.297, 2.330) |
|||||
|
Anthracycline use** |
Yes |
1940 |
2.642 |
<0.001 |
0.755 |
0.377 |
1.571 |
0.001 |
|
(58.01) |
(1.61, 4.33) |
(0.41, 1.41 |
(1.195, 2.064) |
|||||
|
*Wilcoxon rank sum test |
||||||||
|
HSCT, hematopoietic stem cell transplantation; **during 1st 60 days after diagnosis |
||||||||
- Tables 1,3: what do the numbers in the column mean?
; We modified tables. Thank you for your comments.
- Table 2: what do the numbers in the column mean? Different than table 1
; Table 1 is overall demographic characteristics and table 2 is risk factors by age and ON.
- Figure 2: division of patients >20 years in age categories?
; We presented paediatric group and all the patients. Because of there is significant differences in pediatric groups.
- Figure 4: what is the meaning of this figure?
; We added what the blue and red color means in line 186.
Line 186
(Blue line: Paediatric group (2.69±1.77 years), Red line: Adult group (2.33±1.68 years))
- Page 9 lines 237-238: The subset of patients receiving these treatments is exposed to total-body radiation and considerably higher doses of corticosteroids to prevent transplant rejection and combat graft-versus-host disease – is that an assumption or based on patients´data?
; Thank you for your comments. It is confusing that lines 237-238 was assumption or based on our data. Our data showed that in line 236, ‘Our study showed that radiotherapy and HSCT were risk factors in the univariate analysis.’ And line 237-238 were assumption based on previous studies. We modified our description as below
Lines 237-238: The subset of patients receiving these treatments is supposed to be exposed to total-body radiation and considerably higher doses of corticosteroids to prevent transplant rejection and combat graft-versus-host disease.
- Page 9 line 242: Finally, besides age and steroid use, L-ASP was a risk factor in the multivariate analysis (HR=1.99 in all age groups and HR=2.55 in the paediatric group) –significance only in the whole group or pediatric group influencing the whole group? What about young adults?
; Thank you for your comments. At first, we did not consider only young adults but we further analyzed young adults group as the reviewer’s comment. There was no significance in young adults as below and we added this data as supplements table 2 (Line 214).
Supplement Table 2
|
risk factors |
|
|
Univariate |
Multivariate_ON |
Multivariate_Death |
|
|||
|
level |
Median (Q1, Q3) or n (%) |
HR (95% CI) |
p-value* |
HR (95% CI) |
p-value* |
HR (95% CI) |
p-value* |
|
|
|
Age |
29 (20.39) |
0.970 (0.930, 1.012) |
0.162 |
0.972 (0.931, 1.015) |
0.196
|
1.018 (1.003, 1.033) |
0.018 |
|
|
|
Sex |
Male |
912 (60.12) |
1.068 (0.650, 1.755) |
0.795 |
1.000 (0.606, 1.651) |
0.999 |
1.086 (0.911, 1.295) |
0.359 |
|
|
Radiotherapy |
Yes |
521 (34.34) |
1.290 (0.772, 2.154) |
0.331 |
1.019 (0.575, 1.804) |
0.949 |
1.239 (1.012, 1.517) |
0.038 |
|
|
HSCT |
Yes |
833 (54.91) |
1.638 (0.963, 2.787) |
0.069 |
1.324 (0.718, 2.442) |
0.369 |
1.193 (0.965, 1.474) |
0.103 |
|
|
Steroid use** |
Dexamethasone |
567 (37.38) |
2.664 (1.011, 7.017) |
0.047 |
1.846 (0.564, 6.045) |
0.311 |
2.113 (1.407, 4.174) |
<0.001 |
|
|
Prednisone |
354 (23.34) |
2.810 (1.051, 7.508) |
0.039 |
1.687 (0.506, 5.618) |
0.395 |
1.277 (0.831, 1.962) |
0.266 |
|
|
|
Dexamethasone + Prednisone |
408 (26.90) |
2.647 (0.965, 7.263) |
0.059 |
1.620 (0.470, 5.578) |
0.445 |
1.685 (1.099, 2.584) |
0.017 |
|
|
|
Not used |
188 (12.39) |
1 (ref) |
|
1 (ref) |
|
|
|
|
|
|
Asparaginase use |
Yes |
615 (40.54) |
1.783 (1.094, 2.908) |
0.020 |
1.510 (0.862, 2.644) |
0.150 |
1.964 (1.622, 2.377) |
<0.001 |
|
|
Anthracycline Use** |
Yes |
308 (20.30) |
2.316 (1.141, 4.698) |
0.020 |
1.287 (0.514, 3.219) |
0.590 |
1.056 (0.784, 1.423) |
0.720 |
|
- Page 9 lines 245-252: What is the meaning of this sentence regarding ON?
; Thank you for your question. We added this sentence to explain our subjects of study related to age set-up. But the reviewer considered this sentence is unnecessary, and other reviewer commented that ‘The manuscripts could also be shortened’, then we deleted this sentence.
- Page 9 lines 255-256: In our study and other studies, age is the most consistent and significant risk factor, with patients aged ≥10 years at the highest risk across treatment regimens and study groups [8-10,13,53]. – what are the data in young adults? Reference here are pediatric (adolescent) data.
; Thank you so much for your comments. As reviewer’s comment we described only pediatric group and references are pediatric data. About young adult group, there are inconsistent results from previous studies. We clearly modified our description as
In our study and other studies, age is the most consistent and significant risk factor, with patients aged ≥10 years at the highest risk across treatment regimens and study groups in pediatric age.
- Page 9 lines 265-266: in our study, the use of dexamethasone in the paediatric group was lower than that in the prednisone group – what is the meaning of this sentence? Why highlight the difference between dexamethasone and prednisone?
; Thank you for your question. As dexamethasone has known related to the development of ON, reducing dexamethasone was considered as one of factors related to preventing ON. This sentence means that we still use dexamethasone in the pediatric group even though it is lower than prednisone in spite of risk of ON.
- Page 9 lines 271-272: ON occurs more frequently in white patients than in black or Hispanic patients. – What is the purpose of this sentence, when describing Korean patients?
; Thank you for your comments. We described this sentence considering ethnicity from previous studies. But the reviewer considered this sentence is unnecessary, and other reviewer commented that ‘The manuscripts could also be shortened’, then we deleted this sentence.
